# Investigating the Influence of Anaesthesiology for Cancer Resection Surgery on Oncologic Outcomes: The Role of Experimental In Vivo Models

**DOI:** 10.3390/medicina58101380

**Published:** 2022-10-01

**Authors:** Ryan Howle, Aneurin Moorthy, Donal J. Buggy

**Affiliations:** 1Department of Anaesthesiology & Perioperative Medicine, Mater Misericordiae University Hospital, D07 R2WY Dublin, Ireland; 2Department of Anaesthesiology, Mater Private Hospital, Eccles Street, D07 WKW8 Dublin, Ireland; 3School of Medicine, University College Dublin, D04 V1W8 Dublin, Ireland; 4EuroPeriscope, ESA-IC Onco-Anaesthesiology Research Group, B-1000 Brussels, Belgium; 5Outcomes Research, Cleveland Clinic, Cleveland, OH 44195, USA

**Keywords:** cancer, surgery, cancer recurrence, cancer surgery, anaesthesia, in vivo model, animal model

## Abstract

The incidence and societal burden of cancer is increasing globally. Surgery is indicated in the majority of solid tumours, and recent research in the emerging field of onco-anaesthesiology suggests that anaesthetic-analgesic interventions in the perioperative period could potentially influence long-term oncologic outcomes. While prospective, randomised controlled clinical trials are the only research method that can conclusively prove a causal relationship between anaesthetic technique and cancer recurrence, live animal (in vivo) experimental models may more realistically test the biological plausibility of these hypotheses and the mechanisms underpinning them, than limited in vitro modelling. This review outlines the advantages and limitations of available animal models of cancer and how they might be used in perioperative cancer metastasis modelling, including spontaneous or induced tumours, allograft, xenograft, and transgenic tumour models.

## 1. Introduction

In 2020, an estimated 18 million cancer cases were newly diagnosed (excluding nonmelanoma skin cancer), accounting for approximately 10 million cancer related deaths [1]. Surgical resection of solid tumours remains a mainstay of management of >60% tumours because it offers the best chance of cure [2]. Cancer related mortality is rarely caused by the primary tumour itself, but instead results from the metastatic process and consequent organ dysfunction, which accounts for up to 90% of cancer-related deaths [3,4].

The original hypothesis that the anaesthetic technique during primary cancer resection surgery of curative intent might influence the risk of cancer recurrence or later metastasis was first proposed over a decade ago [5]. This included debate around a potential pro-tumorigenic effect of opioids [6]. Subsequently, the question arose whether opioid sparing anaesthesia-analgesia techniques (e.g., regional anaesthesia) and/or Total Intravenous Anaesthesia (TIVA) techniques can reduce the risk of cancer recurrence and improve survival outcomes for primary cancer surgery [7]. 

To ultimately prove these hypotheses, large prospective randomised-controlled clinical trials are required to establish if a causal relationship exists between anaesthetic techniques and the risk of cancer recurrence following primary cancer surgery. However, pre-clinical laboratory models, primarily in vivo animal models, retain an important role in translational cancer research for several reasons [8]. Firstly, when designing a robust, prospective, randomised clinical trial it is recommended that the hypothesis is underpinned by quality laboratory evidence to support the trial’s rationale. Secondly, animal models allow researchers with limited resources to test numerous hypotheses, within a more realistic time frame than may be expected in the human clinical setting. Thirdly, in vivo models allow investigators to study the pharmacodynamic effect of various anaesthetic, analgesic, and perioperative interventions on a whole-organism model of cancer biology, which may in turn generate new hypotheses. Lastly, evidence emerging from clinical trials is a slow process and dependent on several external factors, including: the ability to recruit trial participants, availability of personnel, environmental and equipment resources, and large scale funding [9]. For example, the emergence of the CCOVID-19 pandemic had a profound negative impact on ongoing clinical trials other than COVID-19 associated trials [10]. Therefore, experimental evidence from in vivo models will continue to play an important role in supporting or refuting cancer treatment hypotheses.

The emergence of onco-anaesthesiology as a distinct clinical subspecialty has driven the exploration of translational research utilising animal models of cancer, traditionally undertaken by oncology researchers. Therefore, we aimed to summarise animal models of cancer commonly encountered within in vivo translational cancer research and how they may be applied to ongoing research in onco-anaesthesiology.

Animal models of cancer may be classified in a variety of ways. Most simply, they are either spontaneous or induced and mammalian or non-mammalian. Alternatively, they may be categorized by the method of inducing cancer occurrence. However, spontaneously occurring cancers may occur in genetically engineered animal strains or such genetic-engineering may be induced following exposure to various carcinogens, so there is some cross-over in these descriptions. Non-mammalian animals such as zebrafish benefit from being high-throughput and low-cost, ideal for molecular investigation and chemical screening studies, however, significant phenotypical differences limit their usefulness for translational research so they will not be discussed further [11]. Table 1 summarizes the commonly utilized animal models of cancer, each of which are described below.

## 2. Xenograft Model

A xenograft model (Figure 1) involves the transplantation of cancer cells from one species (e.g., human) into a host animal of a different species (e.g., mouse). This model is immediately constrained because it requires an immunocompromised host animal to prevent immunological rejection of the non-species cancer cells. Transplantation may be ectopic (deposition of cancer cells beneath the skin) or orthotopic (deposition of cancer cells targeted at the organ of interest). Alternatively, cancer cells may be administered intravenously to mimic metastatic spread or the seeding that is thought to occur during solid tumour cancer surgery [12]. Patient-derived xenografts represent an evolution of this approach utilizing transplantation of fresh tumour biopsies obtained directly from patients into immunocompromised mice to create so-called tumour grafts or avatar mice to enable testing and identification of individualized therapies [13,14].

The advantages of xenograft models are that they are relatively inexpensive when using commercially available cancer cell lines and attractive for translational research due to the ability to mimic cancer cell biological traits and the direct evaluation of therapeutic targets in human derived cancer tissue [15,16]. However, several disadvantages arise from the requirement for immunocompromised animals, such as a lack of representative immune response or inflammation, superficial vascularization of the grafts and limited stroma-tumour interactions [15]. Tumour growth rate is variable and often slow, tumour cell composition may not represent the heterogeneity present in the parent cancer and it may not result in metastatic spread, all of which limit the detection of clinically significant metastatic outcomes or falsely increase the perceived efficacy of experimental therapeutic interventions [12,16,17]. The xenograft model also requires quality control because many cell lines have unknown sources or poorly documented receptor expression, and regulatory safeguards are needed to protect researchers from the high communicability risk from handling human cancer tissue [16,18]. Despite these shortcomings, xenograft models have been used extensively to identify potential molecular mechanisms underlying the observed anti-tumour effects of propofol and local anaesthetics as well as pro- and anti-tumour effects following exposure to inhalational anaesthetics [19,20,21,22,23].

## 3. Allograft Model

An allograft model (Figure 1) involves the transplantation of cancer cells between animals of the same species (e.g., mouse cancer cells into another mouse), whereas a syngeneic allograft specifically refers to transplantation of cancer cells between genetically identical animals, which effectively eliminates confounding from inter-species interactions.

The main advantage this has over xenograft models is the ability to evaluate the host animal’s cancer-related immune response when assessing the effect of potential therapeutics. Compared to xenografts, allografts produce larger tumours that metastasize more quickly and reliably, enabling consistency when assessing for clinically relevant endpoints [18]. They do, however, present their own limitations. Firstly, allograft models are often artificial, because mice may not form these cancers spontaneously [24]. They also differ from human cancers in a variety of ways, including mice having more resilient immune systems than humans, evident in both innate and adaptive immune responses and variability in the observed stromal interactions [18].

The preservation of the host immune response underpins why these models are often favoured for investigation in onco-anaesthesiology, where immunomodulation by anaesthetic drugs is one of the most frequently proposed mechanisms to explain how differences in perioperative pharmacotherapy may influence cancer outcomes [5,25,26] A common example from onco-anaesthesiology literature is the 4T1 syngeneic allogenic mouse model of breast cancer that has been used to demonstrate the anti-metastatic effects of systemic lidocaine, in combination with propofol or sevoflurane anaesthesia as well as the pro-metastatic effects of methylprednisolone on cancer progression [27,28,29,30].

## 4. Companion Animals

Testing potential therapeutics on spontaneous cancers that develop in household pets (Figure 1) is an often-underutilized approach that can address some of the limitations encountered in the traditionally favoured mouse models. Cancer occurs in dogs at twice the frequency of humans at an average age of 8.4 years, and their shorter lifespan makes it easier to collect and analyse survival data [31,32] Commonly occurring cancers in dogs include lymphoma, melanoma, and mammary carcinoma and the resulting tumours are the closest clinical and histopathological resemblance to human cancers of any other animal model. This includes some identical tumour oncogenes and tumour suppressor genes involved in promoting its development and progression [15]. To our knowledge, companion animals have not been studied in onco-anaesthesiology research, however, veterinary anaesthesiology could potentially be an avenue for further translational research in this field.

## 5. Transgenic Models

Transgenic models (Figure 2) of cancer require genetic engineering that produces genome mutations via either environmental exposure to carcinogens or genetic editing of fertilized embryos to increase the likelihood of cancer occurrence. These are predominantly performed in mice. Chemical carcinogens include N-butyl-N-(4-hydroxybutyl) nitrosamine and asbestos, but induced mutations occur at random and require high-throughput genome sequencing to identify them and extensive validation to identify the specific role of each mutation [13,15]. Knock-in or knock-out mice can be generated that either promote the expression of various oncogenes such as HRAS in breast cancer, or silence the effect of tumour-suppressor genes such as Brca1 in breast cancer, respectively [33]. Genetic editing may be performed in a variety of ways including retroviral infection, microinjection of DNA (standard transgene approach) or the ‘gene-targeted transgene’ approach. The ‘gene-targeted transgene’ approach involves targeted manipulation of embryonic stem cells where the desired mutation is identified and expanded before reinjection into mouse blastocytes, whereas it is not possible to control the location and pattern of gene expression using traditional methods, which may lead to unexpected results due to effects on neighbouring genes [15]. Novel alternatives include transposon-based insertional mutagenesis or the clustered regularly interspaced short palindromic repeats (CRISPR)/associated (Cas9) engineered nuclease system, which further enhance and/or simplify the ability to target desired genetic mutations [34]. However, generating new genetically engineered mouse models remains costly, labour-intensive, and time-consuming; it often requires multiple generations of mice to achieve the desired pattern of gene expression and some mutations may be lethal to the embryos or cause developmental abnormalities or sterility [35]. This issue may be partially addressed by the generation of conditional knock-in or knock-out mice where further genetic engineering renders the mutation conditional on certain environmental conditions such as the exposure to tetracycline or tamoxifen [34]. Despite these limitations, several colonies of live mouse models of cancer are commercially available and are frequently utilised for cancer research either alone or as donors for transplantation in allogenic models.

Transgenic cancer models may play a significant role in onco-anaesthesiology research as they allow for testing of drug effects on the onset and progression of an expected cancer in immunocompetent animals, such as what has already been performed with morphine, which had no effect on the onset of cancer development but did hasten cancer progression [36]. However, choosing a favourable strain can be one of the most challenging parts of experimental design so a number of electronic databases such as Cancer Models (CaMOD) have been developed to assist in the selection [37].

In conclusion, there are several in vivo animal models of cancer that may be utilized for conducting translational research in onco-anaesthesiology. Whilst xenograft models are attractive for the ability to tailor the cancer cell biology around the human cancer under investigation, a lack of any representative immune response severely hinders its suitability for onco-anaesthesiology research. Transgenic models demonstrate significant promise in providing representative animal cancers and may be used alone or as donors for allogenic models. Multiple transgenic mouse colonies and cancer cell lines are commercially available to enable rapid integration of in vivo mouse models into novel research, however, due care must be taken to ensure the model chosen is most appropriate for the hypothesis under investigation.

## Figures and Tables

**Figure 1 medicina-58-01380-f001:**
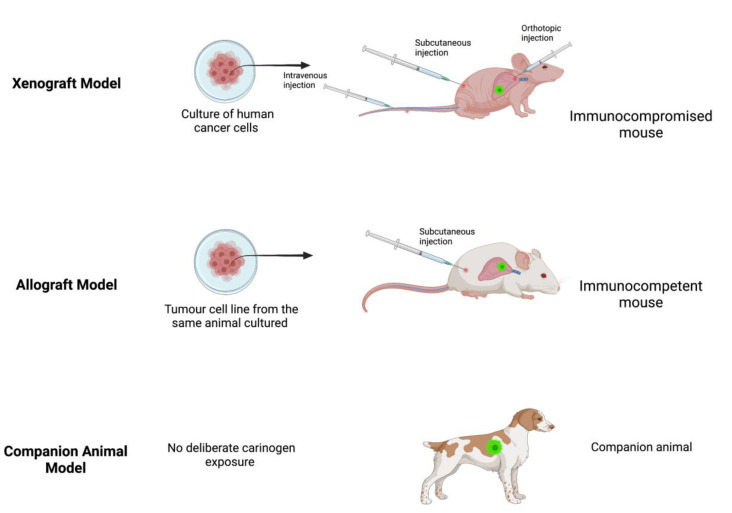
Examples of xenograft, allograft and companion animal models. Created with BioRender.com.

**Figure 2 medicina-58-01380-f002:**
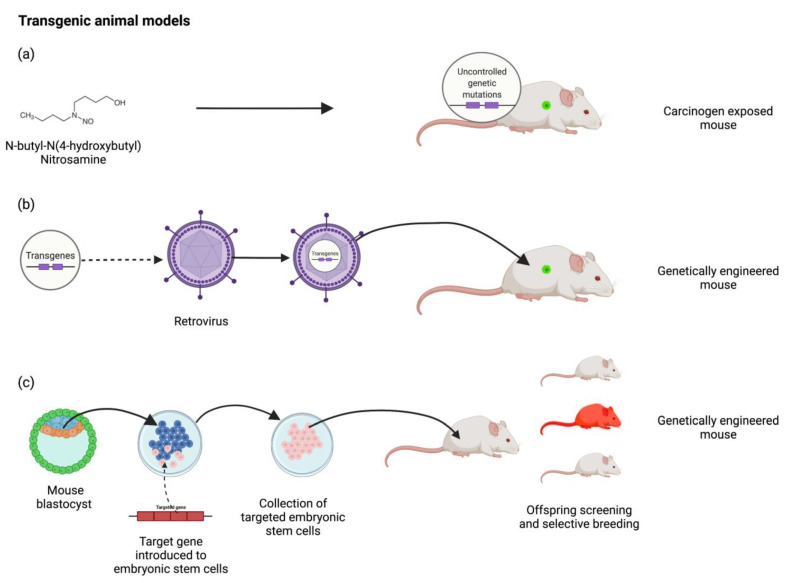
Examples of transgenic animal models. (**a**) Carcinogen exposed mouse (e.g., nitrosamine), (**b**) gentetically engineered mouse by retrovial infection and (**c**) gentetically engineered mouse by gene-targeted transgene approach via genetic manipulation of embryonic stem cells. Created with BioRender.com.

**Table 1 medicina-58-01380-t001:** Animal models of cancer classification.

Classification	Description	Example
Spontaneous companion animals	Spontaneous cancers in household pets	Mammary carcinoma in dogs
Spontaneous Transgenic Knock-in or knock-out cell lines	Genetically engineered animals with specific mutations the precipitate the development of cancer during their normal lifespan	Mice with a rat C3(1) simian virus 40 large tumour antigen fusion gene
Induced Transgenic Carcinogen-exposed Conditional knock-in/knock-out cell lines	Inducing genetic mutation via environmental triggers that precipitate cancer development	N-butyl-N-(4-hydroxybutyl) nitrosamine exposed mice & Tetracycline induced Cre recombinase gene expression system
Induced Allograft Non-syngenic Syngenic	Transplantation of cancer cells between animals of the same species that may (syngenic) or may not (non-syngenic) be genetically identical	4T1 mouse cancer cells transplanted into Bagg Albino (BALB/c) mice
Induced Xenograft Patient derived xenograft (PDX)	Human cancer cells that may be commercially obtained or patient-specific (PDX) transplanted into other animals	Athymic nude mice Severely compromised immunodeficient (SCID) mice

## Data Availability

Not applicable.

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
