# Peer review of "Investigating the Influence of Anaesthesiology for Cancer Resection Surgery on Oncologic Outcomes: The Role of Experimental In Vivo Models"

_medicina, 2022, doi:10.3390/medicina58101380_

Round 1

Reviewer 1 Report

I would like to thank the editor and the journal for inviting us to review this manuscript.

Overall, the topic of this review is interesting and covers academic needs.

However, the amount of data covered in this manuscript and the conclusions drawn from it are not sufficient to constitute an independent scientific research. Therefore, this manuscript is not considered suitable for publication in a peer-review scientific journal until an adequate amount of data has been supplemented and creative conclusions from it have been added.

Other major concerns include:

1) Review studies may highlight reproducibility issues. Therefore, at a minimum, there should be a detailed description of the methodology on which criteria and through which pathways these studies were conceived, and how conclusions will be reached. The manuscript lacks such content.

2) A table or a bibliographic figure for a topic should be presented in a way that summarizes the information according to the characteristics of individual studies so that the reader can easily check what kind of knowledge the information collected in this review creates comprehensively.

3) Staying at the descriptive summary of the collected literature for review is more like an internal report than scientific research. For these, a multi-faceted analysis should be conducted to generate new knowledge and generate valuable hypotheses for subsequent research.

Author Response

We thank these reviewers for their comments. However, there seems to be a fundamental misunderstanding about the objective of this manuscript, which was a narrative review of in vivo methodologies and models of studying the effect of perioperative interventions on cancer. We have no idea what the reviewer means by “creative conclusions” but I suppose it to mean a systematic review and meta-analysis, which this was never intended to be. We have comprehensively discussed relevant available literature around the different in vivo models and simply reject any suggestion that we have covered insufficient material.

It is also stated that “there should be a detailed description of the methodology on which criteria and through which pathways these studies were conceived, and how conclusions will be reached…..” Again, our remit, as directed by Alain Borgeat, who commissioned us to undertake this review, was to provide a narrative overview of in vivo methodologies, which we have delivered, although we are unsure what is intended by ‘pathways (through) which these studies were conceived…”.

Disappointingly, these reviewers suggest that our manuscript is “more like an internal report than scientific research. For these, a multi-faceted analysis should be conducted to generate new knowledge…”. Narrative reviews of this type which we were commissioned to undertake are a key component of the literature and are never designed to create new knowledge, rather to summarise existing knowledge in the field.

Reviewer 2 Report

The authors aimed to review animal models commonly used in onco-anaesthesiology, research, specificaly those trying to verify if a causal relationship exists between anesthetic techniques used in primary cancer surgery and the risk of cancer recurrence. They advocate the use of in vivo animal models instead of those in humans, because of the difficulties associated with large prospective randomised-controlled clinical trials.

The revision is short but sharp in presenting the most common mammalian models used (mainly: xenograph, allograft and transgenic) but I miss reports of the achievements obtained with those models, regarding the main topic: do anaesthetics (themselves  or the anaesthesia technique employed) influence cancer metastasis?

I also wondered if other (non-mammalian) animal models - such as zebrafish and Caenorhabditis elegans – should not be considered for such revision.  

Minor point: language issues: 

“…to study the pharmacodynamic effect of various anaesthesia (ANAESTHETIC?), analgesic and perioperative interventions”

Author Response

We thank the reviewer for their comments and recommendations and hope our updated manuscript will adequately meet their expectations. Taking each point in turn:

  • Regarding the achievements of these models, it was not within our remit to provide a comprehensive review of the onco-anaesthesia literature, but we have made significant changes to address this. Each section now provides examples of how the model has been used including the outcomes of these studies.
  • We have now included reference to non-mammalian models and thank you for your comment regarding this. To our knowledge, these models have not yet been utilised in onco-anaesthesiology translational research.
  • Finally, thank you for highlighting the grammatical error, which has been rectified.

Round 2

Reviewer 2 Report

In this revised version the authors have sufficiently addressed the points raised before.